# Three-Dimensional Accuracy and Stability of Personalized Implants in Orthognathic Surgery: A Systematic Review and a Meta-Analysis

**DOI:** 10.3390/jpm13010125

**Published:** 2023-01-07

**Authors:** Alexandru Diaconu, Michael Boelstoft Holte, Gabriele Berg-Beckhoff, Else Marie Pinholt

**Affiliations:** 13D Lab Denmark, Department of Oral and Maxillofacial Surgery, University Hospital of Southern Denmark, Finsensgade 35, 6700 Esbjerg, Denmark; 2Department of Regional Health Research, Faculty of Health Sciences, University of Southern Denmark, Finsensgade 35, 6700 Esbjerg, Denmark; 3Department of Public Health, Faculty of Health Sciences, University of Southern Denmark, Degnevej 14, 6705 Esbjerg, Denmark

**Keywords:** patient-specific implants, orthognathic surgery, accuracy, stability, three-dimensional, systematic review, meta-analysis

## Abstract

This systematic review aimed to determine the accuracy/stability of patient-specific osteosynthesis (PSI) in orthognathic surgery according to three-dimensional (3D) outcome analysis and in comparison to conventional osteosynthesis and computer-aided designed and manufactured (CAD/CAM) splints or wafers. The PRISMA guidelines were followed and six academic databases and Google Scholar were searched. Records reporting 3D accuracy/stability measurements of bony segments fixated with PSI were included. Of 485 initial records, 21 met the eligibility (566 subjects), nine of which also qualified for a meta-analysis (164 subjects). Six studies had a high risk of bias (29%), and the rest were of low or moderate risk. Procedures comprised either single-piece or segmental Le Fort I and/or mandibular osteotomy and/or genioplasty. A stratified meta-analysis including 115 subjects with single-piece Le Fort I PSI showed that the largest absolute mean deviations were 0.5 mm antero-posteriorly and 0.65° in pitch. PSIs were up to 0.85 mm and 2.35° more accurate than conventional osteosynthesis with CAD/CAM splint or wafer (*p* < 0.0001). However, the clinical relevance of the improved accuracy has not been shown. The literature on PSI for multi-piece Le Fort I, mandibular osteotomies and genioplasty procedure is characterized by high methodological heterogeneity and a lack of randomized controlled trials. The literature is lacking on the 3D stability of bony segments fixated with PSI.

## 1. Introduction

Titanium patient-specific fixation plates (PSI) are becoming more frequently used in orthognathic surgery (OS), due to accurate positioning and fixation for osteosynthesis and the supposedly resultant reduced operation time [1,2,3,4,5]. Nevertheless, the challenge of PSI usage lies in the expertise and labor required in their design manufacturing for each orthognathic case, thereby adding to operative costs [1,3,4,5]. In order to justify their use, the literature on the accuracy of PSIs, especially in comparison with the less expensive computer-aided design/computer-aided manufacturing (CAD/CAM) inter-occlusal wafers, has been developed in recent years. Studies have highlighted the use of PSI in OS for mono- and bimaxillary procedures, as well as for genioplasty, in combination with a three-dimensional (3D) virtual surgical planning (VSP) workflow [1,2,3,4,5]. However, due to the heterogeneous methodology and a lack of controlled trials in the literature [1,2], the most recent developments have yet to offer a precise recommendation on the use of PSI in OS. 

Standardization in measurements of craniofacial surgery’s accuracy and stability, and skeletal movement at follow-up longer than one year postoperatively, creates challenges, an issue raised by Zavattero et al. [6]. Computer-assisted surgical assessment using voxel-based registration (VBR) [7,8,9] is superior to the traditional technique for the assessment of the surgical outcome [10], offering accurate superimposition methods for measurements and comparisons [7,8,9]. Virtual surgical planning is an accurate approach to OS [11], creating virtual results with deviations less than 2 mm/4°, which are considered clinically acceptable [12,13,14,15,16,17,18]—although standardization lacks consensus [15,19]. A standardized protocol for accuracy and stability measurements in OS implying VBR and a validated automated analysis of outcomes in 3D has already been established [20] and implemented in the literature [21,22,23,24]. The purpose of this study was, therefore, to perform a literature review evaluating whether the use of PSI in OS is examined for accuracy and stability by established 3D methods such as VBR and a validated automated analysis of outcomes in 3D—and compared to the use of CAD/CAM splints. The specific aim of this systematic review was to answer the following research question: Is PSI accurate/stable in OS, and more so than the use of CAD/CAM splints, as evaluated by the established standardized 3D methods [20,21,22,23,24]?

## 2. Materials and Methods

### 2.1. Study Design

For this review, the authors followed the criteria established in the Preferred Reporting Items for Systematic Reviews and Meta Analyses (PRISMA) guidelines [25].

The predictor variable was the use of PSI in OS, and the outcome variables were the translational (antero-posterior, transversal and vertical) and rotational (pitch, roll, yaw) accuracy and stability measurements performed by the standardized 3D methods [20,21,22,23,24].

### 2.2. Eligibility Criteria

Inclusion criteria: (1) studies on humans undergoing (2) corrective orthognathic procedures employing the use of Le Fort I (either single-piece or segmental) and/or mandibular osteotomy (BSSO, Inverted-L or vertical) and/or genioplasty procedures with (3) patient-specific titanium fixation plates, and (4) measurements of accuracy of the bone movements in 3D must be present.

Exclusion criteria: in vitro studies, studies including patients under the age of 14, procedures of reconstructive surgery (e.g., trauma, oncology-related, temporomandibular joint replacement, mandibular free flap reconstruction), studies not in the English language, reviews that contain already identified records (excluded only after full text consideration), case reports of a single patient, abstracts and letters.

### 2.3. Search Strategy

A systematic electronic search was conducted on the following databases: Cochrane Library, Embase (Ovid), Medline (Ovid), PubMed, Scopus and Web of Science. Additionally, a hand search was performed on Google Scholar. The search strings can be found in Table A1 (Appendix A). Furthermore, the reference lists of the selected records were hand-searched for potentially omitted relevant entries. All records were managed using EndNote X9 (Clarivate, Philadelphia, PA, USA).

### 2.4. Data Collection

The author, year published, study design, number of patients, demographic data, treatment, location of PSI, accuracy measurements and software used were recorded. The records were evaluated by the lead authors (A.D. and M.H.), noting weaknesses in study design and analysis. The senior author (E.P.) independently reviewed the potential records, and the final selection of included studies is the result of this process.

### 2.5. Risk of Bias Assessment

The studies that were non-randomized were assessed for risk of bias using the methodological index for non-randomized studies (MINORS) [26]. The MINORS system scores individual studies as “not reported” (0 points), “reported but inadequate” (1 point) or “reported and adequate” (2 points). Final scores in the ideal scenario would be 16 points for non-comparative studies and 24 points for comparative studies.

The studies that were randomized were assessed using the Cochrane Collaboration’s tool for assessing the risk of bias in randomized trials [27], using the following quality criteria: random sequence generation, allocation concealment, blinding of participants, blinding of outcome, incomplete data addressed and selective reporting. The possible assessments were a low, high or unclear risk of bias, respectively. 

### 2.6. Meta-Analysis

Where permitted by the resemblance and homogeneity of the study design and measured outcomes of the records, a meta-analysis was conducted using STATA version 17.0 (StataCorp, College Station, TX, USA). In the cases where the mean value was not provided, the median was used instead. After checking for normal distribution, the standard deviation was calculated using the interquartile range [28]. A random effect model was used for the overall estimate to consider potential variation in the data. 

## 3. Results

### 3.1. Liteature Search

The results of the literature search performed on 14 July 2022 and updated in October 2022 are summarized in Figure 1. The database searches yielded 485 results, with an additional five records identified through manual searching. After deduplication, 142 records were removed, leaving 348 records for title and abstract screening. After screening the titles and abstracts, 294 irrelevant records were removed, leaving 54 records for full text assessment. Thirty-three full text records were excluded, as they did not meet the eligibility criteria. The remaining 21 studies were included in the analysis. The references of these studies were appraised (552 titles) and no additional articles were selected.

### 3.2. Characteristics of Eligible Studies

All included studies were published between 2017 and 2022. A total of 566 patients from 21 articles were included for this systematic review. Eleven articles had a prospective study design [13,29,30,31,32,33,34,35,36,37,38], of which five were randomized [31,34,36,37,38], four were clinical trials [31,34,36,38] and two were multi-center [34,38]. The study characteristics are presented in Table 1.

The number of patients ranged from 4 to 82 [39,40]. Two of the records used the same cohort as part of a study performing different accuracy measurements [29,30]. Another two records consisted of different time-points of the same study [34,38], where a subset of the cohort had the measurements of the first time-point reported again for stability assessment [38]. These patients have only been included once in the total count for this review. 

Sixteen records evaluated the accuracy of PSI for the maxilla [13,31,32,33,34,36,37,38,39,40,41,42,43,44,45,46], out of which eight exclusively focused on non-segmental Le Fort I procedures [31,32,34,36,38,43,45,46] and two on non-segmental Le Fort I procedures in conjunction with the accuracy of PSI for mandibular osteotomies [13,39]. Six studies included mixed single and segmental Le Fort I osteotomies [33,37,40,41,42]. One of the included studies investigated exclusively segmental Le Fort I osteotomies [44]. Three records reported the accuracy of PSI for mandibular osteotomies exclusively [29,30,47]. Two studies investigated the accuracy of PSI for genioplasty. Table 2 contains the accuracy measurements and Figure 2 contains a Venn diagram of the study distribution.

### 3.3. Risk of Bias

MINORS scores ranged between 8 and 12 for the non-comparative studies, showing a moderate risk of bias (mean: 10 points), and they were 16–22 for the comparative studies, showing a low risk of bias (mean: 17 points) (Table 3).

The risk of bias assessment for randomized trials indicated an overall low risk of bias. In only one case, there was a high risk of attrition bias due to the previously established sample size not being achieved at the outcome reported one year postoperatively [38] (Table 4 and Appendix B).

### 3.4. Meta-Analysis: PSI Accuracy in Single Le Fort I Osteotomy Procedures

The studies were divided into categories based on the PSI location: at the Le Fort I level, mandible or genioplasty. The only subset that had sufficient data for a meta-analysis comprised those studies employing single Le Fort I osteotomies exclusively. The studies reporting absolute values were analyzed [32,34,36,39,43,45]. A stratification of the meta-analysis on absolute and signed values was done to allow correct interpretation of the estimates on two studies [13,46]. Furthermore, to allow a similar presentation of all values in one estimate, one stratum was used for a study that provided a very narrow standard deviation that was an outlier among other studies [31].

The outcome variables and the antero-posterior, transversal and vertical error measurements were registered. An analysis was also conducted on the subset of studies that included pitch, roll and yaw error measurements [34,36,39,43,45,46]. This analysis was also stratified, with a stratum for the studies reporting signed values separate from the rest.

The main results for the analysis with absolute values are presented in the following and shown in Table 5, Table 6, Table 7 and Table 8. The absolute values are considered the most relevant results, because signed results are not representative of the magnitude of errors of bony movements. However, the signed results are representative of the direction of the errors.

The mean (standard deviation) absolute translational deviation was 0.30 (0.18) mm laterally, 0.50 (0.13) mm antero-posteriorly and 0.42 (0.14) mm vertically. The mean absolute rotational deviation was 0.65° (0.35) for pitch, 0.36° (0.17) for roll and 0.26° (0.12) for yaw. All translation and rotation deviations were statistically significant (*p* < 0.0001). Figure 3, Figure 4, Figure 5, Figure 6, Figure 7, Figure 8, Figure 9, Figure 10, Figure 11, Figure 12, Figure 13 and Figure 14 show the meta-analysis forest plots.

A meta-analysis was also possible on four records containing comparison data between procedures with PSI and a control group undergoing a CAD/CAM splint or wafer-based procedure [31,34,35,36,45]. This was also presented in a stratified manner, as it included the study by Hanafy et al., with a very narrow standard deviation [31]. The average translational difference between the PSI and the control was −0.85 mm laterally, −0.68 mm antero-posteriorly and −0.39 mm vertically; the average rotational difference was −2.35° for pitch, −0.47° for roll and −0.56° for yaw. The negative differences indicate that Le Fort I osteotomies with PSI deviate less from the planned bone position than procedures using CAD/CAM splints or wafers. All differences were statistically significant (*p* < 0.0001).

All of these results were considered acceptably below the thresholds of clinical relevance of 2 mm and 4°.

Some of the included records could not be included in the meta-analysis due to the following reasons: the relevant data from one study were redundant [38] because it originated in a cohort that was already included in the meta-analysis [34]; five records had mixed results from single and segmental Le Fort I procedures [33,37,40,41,42].

### 3.5. Mixed Le Fort I Procedures

Five studies reported the PSI accuracy on cohorts containing mixed single and segmental Le Fort I osteotomies [33,37,40,41,42], respectively. The main findings of these studies were below the clinically relevant thresholds of 2 mm of translational deviation and 4° rotational deviation. Individually, they indicated that PSI procedures were at least as accurate as splint-based procedures, and, in the case of particular landmarks, more accurate. However, due to heterogeneity in methodology, a meta-analysis could not be performed on this group.

Karanxha et al. compared PSI with a control group using CAD/CAM splints, and the PSI group was found to be more accurate in the translational movement at the A point (*p* = 0.008) and anterior nasal spine (ANS) (*p* = 0.045), and in the rotational movement in roll (*p* = 0.04) and yaw (*p* = 0.04) [33]. Furthermore, the PSI procedures were found to be accurate, with no statistically significant differences in either translational or rotational movements between the actual outcome and the planned result [33].

Jones et al. reported, on a cohort of 82 patients, that repositioning of the maxilla with PSI versus CAD/CAM splints resulted in a smaller mean error for PSI, which was statistically significant for lateral and vertical movements and incisor angulation [40]. Furthermore, statistically significant differences were reported for the sagittal movement of the UI between cranial and caudal repositioning of the maxillae in the PSI group, with upward movement affecting the sagittal position of the UI less (*p* = 0.02) [40].

Sanchez-Jauregui et al. reported significant differences in the antero-posterior and vertical accuracy, with PSI procedures being more accurate than CAD/CAM splint procedures (*p* < 0.05) [37].

Abel et al. reported discrepancies below 0.6 mm in all dimensions at the A point when comparing the planned result with the outcome achieved with PSI on maxillae, which were considered statistically significant against a threshold of 2 mm, rendering PSI procedures highly accurate [41].

### 3.6. Segmental Le Fort I

Rios et al. (2022) reported accuracy levels below the clinically relevant 2 mm deviation between the planned and postoperative result using PSI on a cohort of segmental Le Fort I maxillae in all three dimensions. Furthermore, no statistically significant differences were found in the average discrepancies between the lateral, antero-posterior and vertical axes [44].

### 3.7. Mandibular Osteotomy Procedures

Six of the included records investigated the accuracy of PSI in mandibular osteotomy procedures [13,29,30,33,41,47]: three reported exclusively mandibular PSI accuracy measurements [29,30,47], and three of them as part of procedures applying PSI to both the maxilla and the mandible [13,33,41]. Abel et al. [41] studied mandible-first and maxilla-first procedures separately, and reported mean errors below 1.15 mm at the B point and below 1.29 mm at the pogonion (*p* < 0.01), with no significant differences with respect to the sequencing. Karanxha et al. [33] concluded that PSI yields more accurate transfer results than the conventional method, with significant differences at the B point and both lower canines (*p* = 0.049) for translation and no significant differences for rotation. Li et al. [13] reported deviations for the mandibular dental arch, mandibular body and proximal segments separately. All reported vales were below 2 mm and 4°, highlighting the PSI as accurate, but no statistical test assessed the significance of the discrepancies.

In two studies, Badiali et al. [29,30] reported separately on the tooth bearing fragment of the mandible, the Rami and plates and found that all discrepancies were below the clinically significant thresholds of 2 mm and 4°.

### 3.8. Genioplasty Procedures

Li et al. and Ruckschloss et al. found PSI for genioplasty accurate and within the clinical limits of 2 mm and 4° of error [35,48]. However, according to Ruckschloss et al. [48], the aesthetic outcomes, including the soft tissue, are not evidently quantifiable; thus, no conclusion could be drawn on the superiority of PSI over conventional osteosynthesis for genioplasty.

### 3.9. Stability

Van der Wel et al. analyzed the discrepancy in bone movements with PSI one year postoperatively and concluded that there is no clinically relevant difference in the use of PSI or conventional osteosynthesis for OS on single Le Fort I maxillae [38]: both methods yielded relapse lower than 1 mm and 1 degree.

## 4. Discussion

The purpose of this review was to investigate in the literature whether the use of PSI in OS is accurate and stable, and more so than the use of CAD/CAM splints. Not all studies were homogenous in their methodology. As a result, they were grouped. The most homogenous grouping comprised records measuring the translational and rotational accuracy of one-piece Le Fort I maxillae. Hence, a meta-analysis was performed on this group.

According to the meta-analysis, PSIs are accurate within the clinically acceptable thresholds of 2 mm/4° in the use of OS on single-piece Le Fort I maxillae, and more accurate than a conventional osteosynthesis and CAD/CAM splint workflow. According to the qualitative synthesis, the use of PSI in OS on segmental le Fort I procedures, mandibular osteotomies and genioplasties is accurate within the clinically acceptable thresholds of 2 mm/4°. However, the heterogeneity in methodology and a lack of prospective randomized clinical trials prevent a definitive conclusion. The stability of PSI in OS has been sparsely investigated, with a single study concluding that PSI use on nonsegmental Le Fort I maxilla fixation is comparable to conventional fixation in terms of stability [38].

The studies investigating PSIs’ accuracy in genioplasty procedures and mandibular osteotomies were not sufficient in magnitude and in homogeneity to be fit for a meta-analysis. Further studies are required. However, a qualitative synthesis indicates discrepancies below 2 mm and 4° for the existing studies.

One randomized controlled trial was available on the stability of PSI on non-segmental Le Fort I maxillae. The novel findings indicate that PSI is stable and comparable with conventional fixation methods [38]. The results of the study are limited only by the sample size, and they provide a first view in an area where the literature is lacking.

The risk of bias assessments reveal a wide range of bias levels, pertaining especially to non-randomized, retrospective study designs with a lack of control groups. In the case of stability, the single long-term follow-up study fitting the criteria of the systematic review, although providing novel data, had a high risk of attrition bias, leading to skewness in group sizes.

To the authors’ knowledge, this is the first meta-analysis of studies reporting complete 3D measurements of PSI accuracy of maxillae. The results corroborate previous reviews that there is a need for high-quality studies on the accuracy and stability of PSI in OS [1,2,5]. Although the literature on the topic has accelerated recently, a complete meta-analysis on the accuracy of PSI for all maxillary and mandibular segments is still infeasible due to the heterogeneous methodology and lack of randomized clinical trials. Additionally, the recommended standardized method for the 3D accuracy and stability assessment of OS is not employed in the majority of these works, and neither are the processes automated. Lastly, the prospective calculation of the sample size is sparsely performed.

There are several limitations to the presented meta-analysis. Within the included records, translational error measurements are performed at the upper incisors, at the geometrical center of the bone segment, or are derived from a rigid transformation matrix of the bone segment. These records have been pooled together for the analysis while acknowledging the potential weakness of not using the same exact anatomical landmarks. It is considered less informative to have further stratified the analysis based on this difference in the specificity of the measurement point, as the analysis of such reduced groups would prevent any meaningful conclusions from being drawn. Furthermore, some of the mean values required in the analysis were estimated from the available median values and interquartile range values. Lastly, the stratum of the meta-analysis with controlled studies contains only five records, and it is further stratified by one of the studies reporting an unusually narrow standard deviation. Despite these potential limitations, the heterogeneity estimates (I^2^) of the chosen groups were low.

Regarding the issue of the standardization of accuracy measurements in OS, the literature seems to move towards the full 3D measurement of translation and rotation. However, heterogeneity persists in methodologies with respect to the points/landmarks at which measurements are performed. Future research is required to focus on PSI accuracy in OS, in regard to multi-piece Le Fort I procedures, mandibular osteotomies, genioplasty and long-term stability measurements, with a prospective study design and control groups. To ensure high-quality results and comparability, future researchers are also encouraged to focus on automatic 3D analysis according to standardized methods [20,21,22,23,24].

It should be noted that surface-based registration (SBR) provides an alternative method to VBR for the 3D assessment of the surgical accuracy and stability of orthognathic surgery, which has been applied in several of the included studies in the present systematic literature review [13,29,30,31,32,35,36,40,43,45,46,47,48]. Surface-based registration aligns two 3D surfaces using the iterative closest point algorithm [49], and SBR has been shown to be a reproducible method [50,51]. Both registration methods have been found to be reliable and accurate [52,53,54]. Although VBR has been shown to be more consistent and efficient than SBR, the differences between the two methods were statistically insignificant [52,53,54]. Another comparative study proved that SBR was more accurate and reliable than VBR on the mandibular ramus for the long-term 3D assessment of condylar remodeling following orthognathic surgery [55]. However, it was concluded that the performance difference might have been caused by the application of an inappropriate reference structure proposed in the literature [55,56]. Hence, according to the literature, no statistically significant or clinically relevant differences have yet been shown in the performance of the two methods. Consequently, both methods have been found to be applicable for the 3D assessment of the surgical accuracy and stability of orthognathic surgery.

## 5. Conclusions

The purpose of this systematic review was to determine whether PSIs are accurate and stable, and more so than conventional CAD/CAM-based wafers and conventional osteosynthesis in OS. The results of the meta-analysis indicate that single-piece Le Fort I osteotomy outcomes are accurate in 3D when compared to the planned movements, falling within the clinically acceptable thresholds of 2 mm and 4°. PSIs are more accurate than the conventional repositioning and fixation methods for single-piece Le Fort I repositioning, with statistical significance favoring PSIs in all three dimensions of translation and rotation. However, the clinical relevance of this improvement has not been shown. Statistical analysis of PSI accuracy for segmental Le Fort I osteotomies, mandibular osteotomies and genioplasty was not possible due to heterogeneity in the literature. However, a qualitative synthesis indicates that PSIs are a viable and accurate method for these procedures in OS. The results are comparable to or better than the conventional methods, but further research is required in the form of randomized controlled trials in order to draw definitive conclusions. The literature on the 3D stability of PSI in OS is sparse, with one novel randomized controlled trial indicating that PSI use in segmental Le Fort I osteotomies provides comparable stability to conventional fixation. Further research is required on the stability of PSI use in OS.

## Figures and Tables

**Figure 1 jpm-13-00125-f001:**
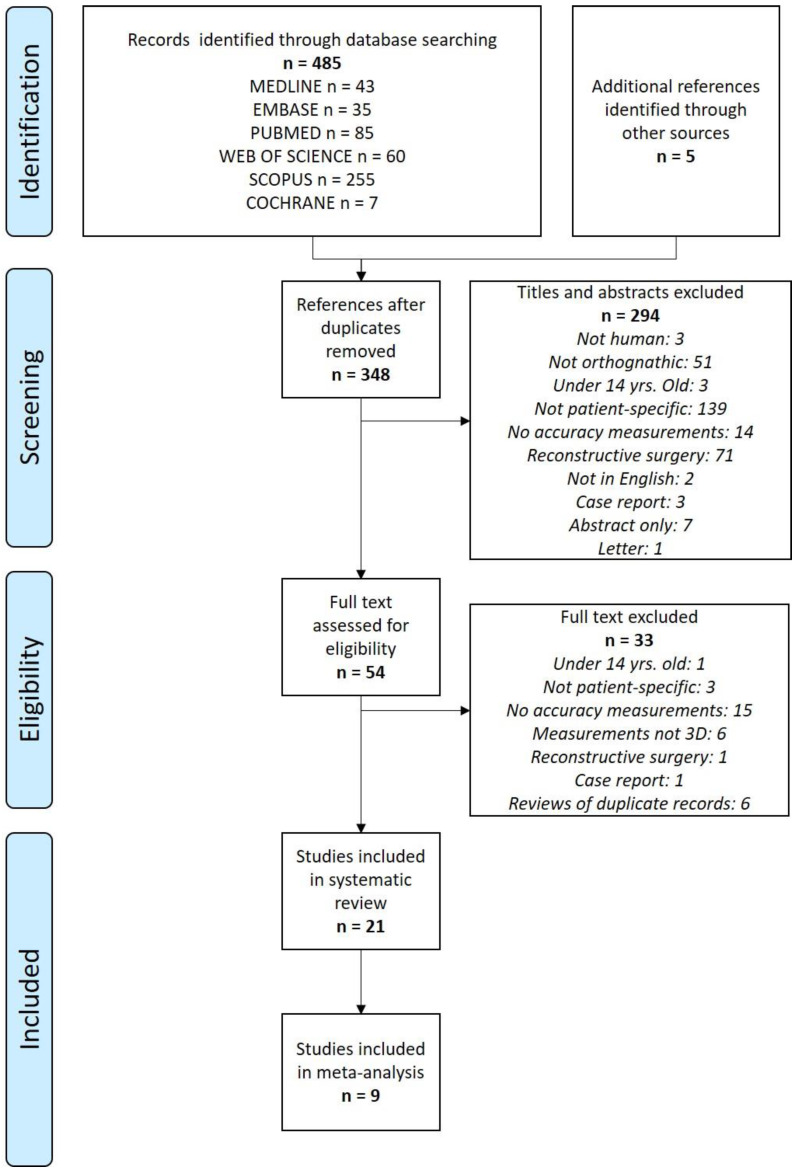
PRISMA flow diagram.

**Figure 2 jpm-13-00125-f002:**
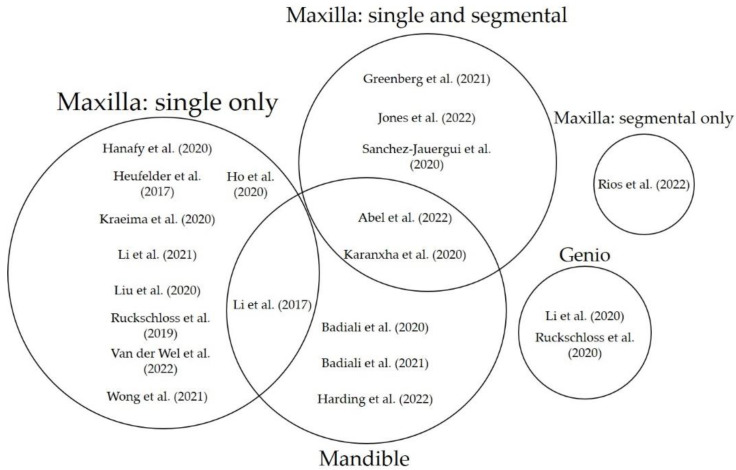
Venn diagram of anatomical location of PSI in the included records with 3D assessment.

**Figure 3 jpm-13-00125-f003:**
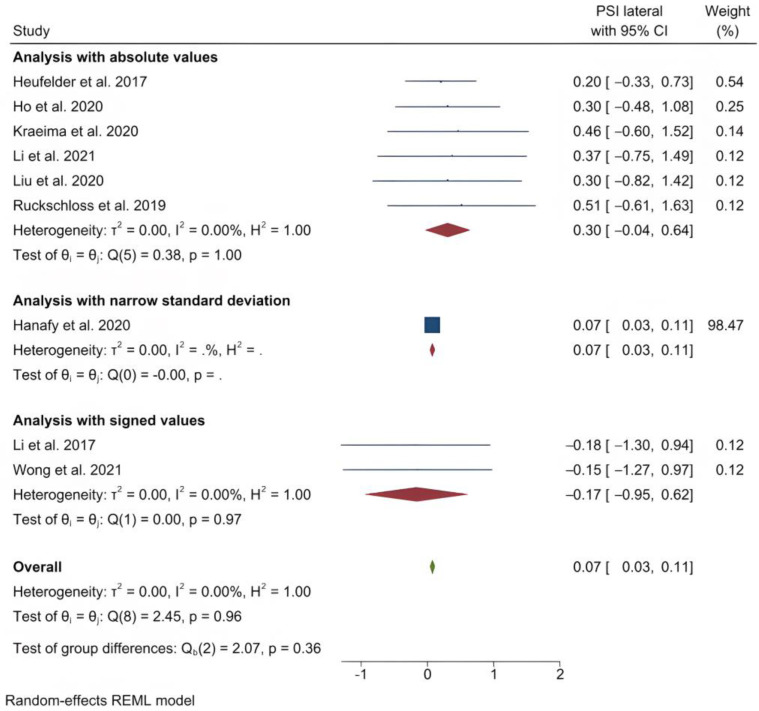
Forest plot showing mean lateral deviation.

**Figure 4 jpm-13-00125-f004:**
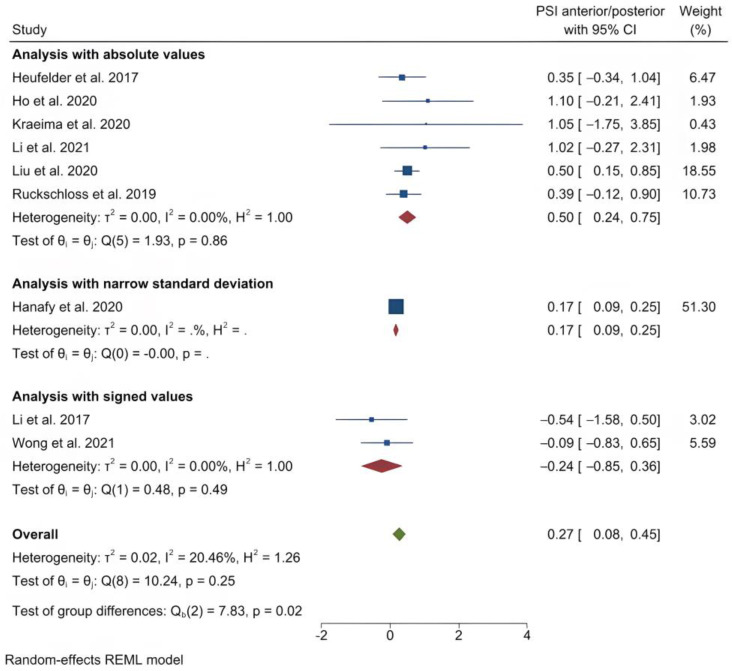
Forest plot showing mean antero-posterior deviation.

**Figure 5 jpm-13-00125-f005:**
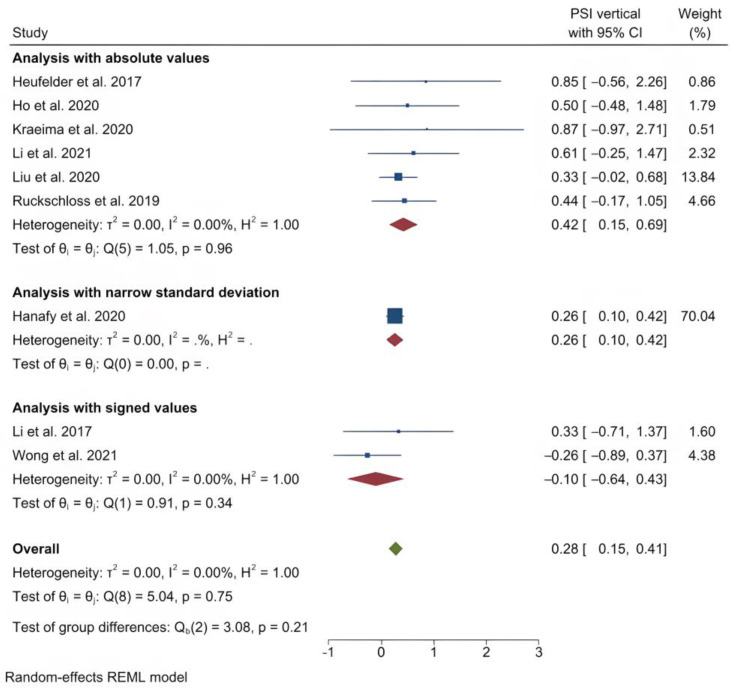
Forest plot showing mean vertical deviation.

**Figure 6 jpm-13-00125-f006:**
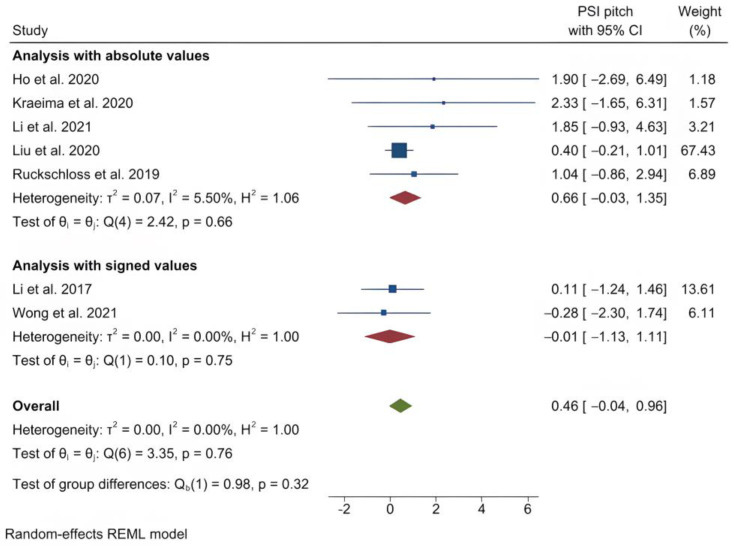
Forest plot showing mean pitch deviation.

**Figure 7 jpm-13-00125-f007:**
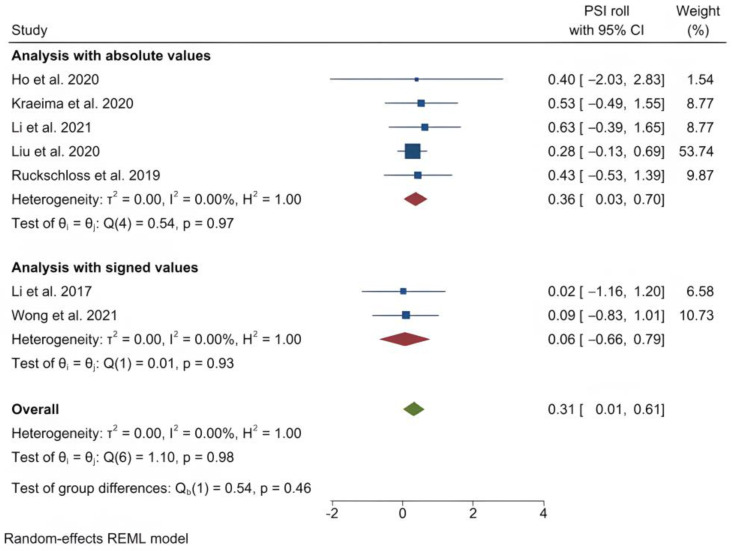
Forest plot showing mean roll deviation.

**Figure 8 jpm-13-00125-f008:**
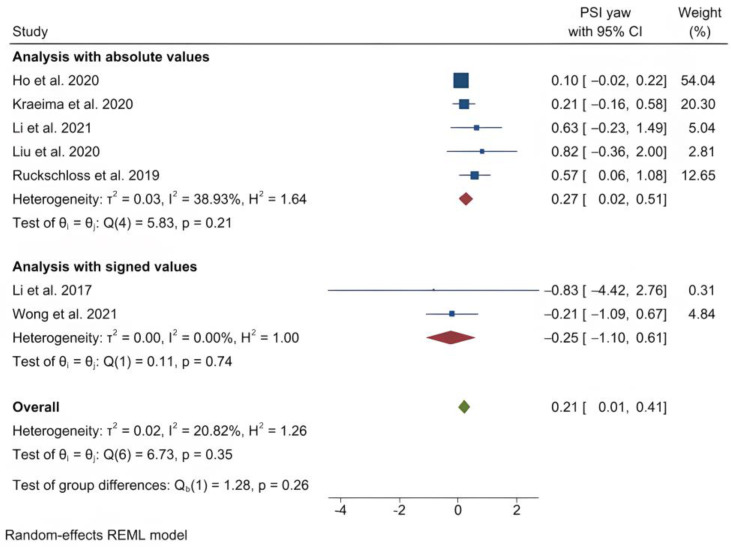
Forest plot showing mean yaw deviation.

**Figure 9 jpm-13-00125-f009:**
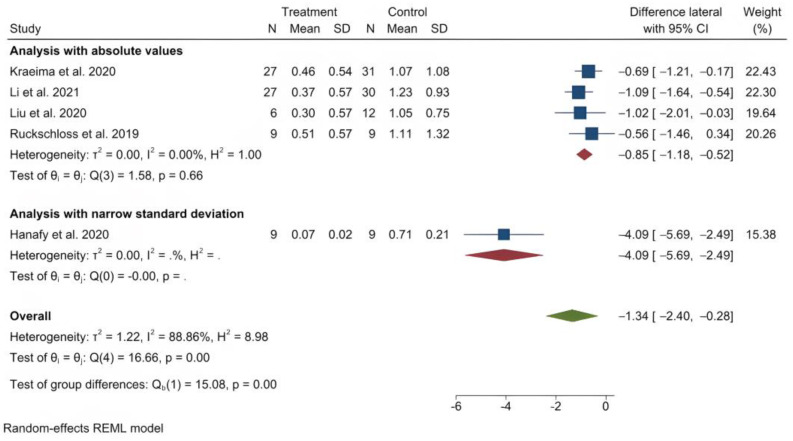
Forest plot showing differences in lateral deviation between PSI and control (conventional osteosynthesis and CAD/CAM splint or wafer).

**Figure 10 jpm-13-00125-f010:**
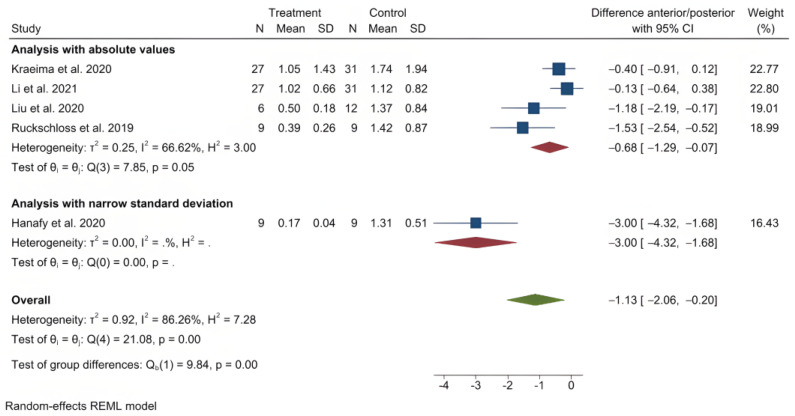
Forest plot showing differences in antero-posterior deviation between PSI and control (conventional osteosynthesis and CAD/CAM splint or wafer).

**Figure 11 jpm-13-00125-f011:**
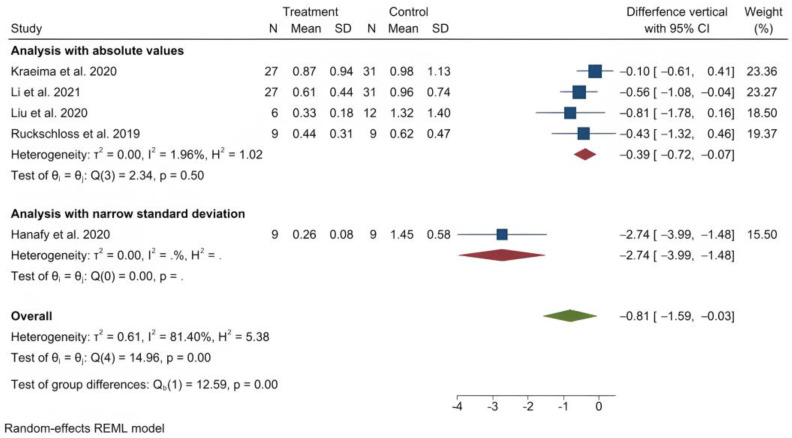
Forest plot showing differences in vertical deviation between PSI and control (conventional osteosynthesis and CAD/CAM splint or wafer).

**Figure 12 jpm-13-00125-f012:**
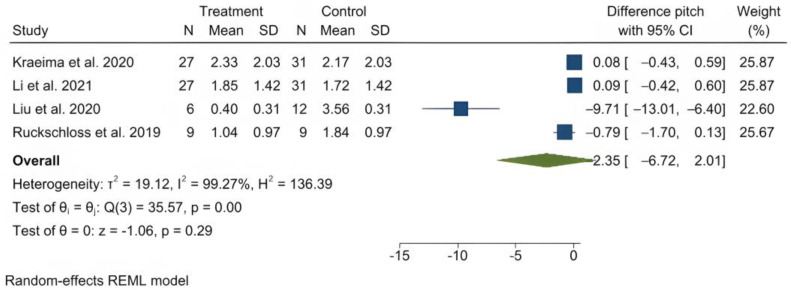
Forest plot showing differences in pitch deviation between PSI and control (conventional osteosynthesis and CAD/CAM splint or wafer).

**Figure 13 jpm-13-00125-f013:**
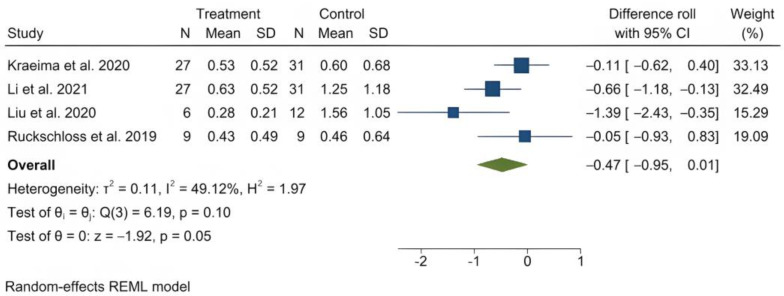
Forest plot showing differences in roll deviation between PSI and control (conventional osteosynthesis and CAD/CAM splint or wafer).

**Figure 14 jpm-13-00125-f014:**
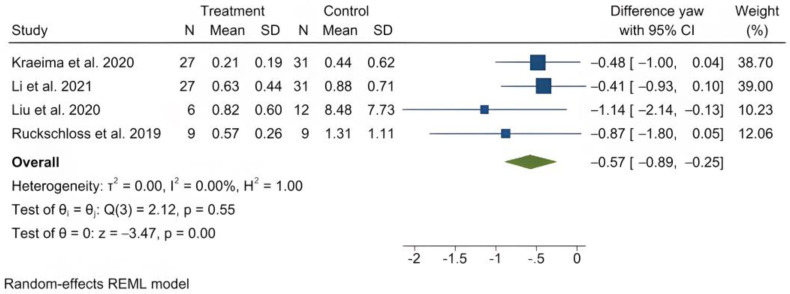
Forest plot showing differences in yaw deviation between PSI and control (conventional osteosynthesis and CAD/CAM splint or wafer).

**Table 1 jpm-13-00125-t001:** Summary of records.

Author, Year	Study Design	N	Mean Age (Range)	Treatment	PSI Component
Abel et al. (2022)	Cohort (retrospective)	49	F: 24 M: 25.6	Bimaxillary osteotomy	Maxilla (single or split) and mandible
Badiali et al. (2021)	Cohort (prospective)	22 *	26 (18–43)	Bimaxillary osteotomy	Mandible
Badiali et al. (2020)	Cohort (prospective)	22	26 (18–43)	Bimaxillary osteotomy	Mandible
Greenberg et al. (2021)	Series (retrospective)	10	25.7	Bimaxillary osteotomy	Maxilla (single or segmental)
Hanafy et al. (2020)	RCT (prospective)	18	21.2 (19–24)	Bimaxillary osteotomy	Maxilla (single)
Harding et al. (2022)	Cohort (retrospective)	55	PSI: 28.0 (16–52) Splint: 25.9 (16–47)	Bimaxillary osteotomy	Mandible
Heufelder et al. (2017)	Series (prospective)	22	NS (17–59)	Bimaxillary osteotomy	Maxilla (single)
Ho et al. (2020)	Cohort (retrospective)	4	55 (48–62)	Bimaxillary osteotomy bilateral sagittal split osteotomy	Maxilla (single)
Jones et al. (2022)	Cohort (retrospective)	82	23.2 (14–54)	Bimaxillary osteotomy	Maxilla (single or segmental)
Karanxha et al. (2021)	Cohort (prospective)	16	26.4	Bimaxillary osteotomy	Maxilla (single or segmental) and mandible
Kraeima et al. (2020)	RCMCT (prospective)	58	27.6 (19–60)	Le Fort I osteotomy	Maxilla (single)
Li et al. (2017)	Series (prospective)	10	NS (18–27)	Bimaxillary osteotomy	Maxilla (single) and mandible
Li et al. (2020)	Cohort (prospective)	15	NS (18–30)	Genioplasty, with or without simultaneous Le Fort I and/or bilateral sagittal split osteotomy	Chin
Li et al. (2021)	RCT (prospective)	58	PSI: 23.8 (19–32) Splint: 23.6 (19–33)	Bimaxillary osteotomy	Maxilla (single)
Liu et al. (2020)	Cohort (retrospective)	18	NS (17–30)	Bimaxillary osteotomy	Maxilla (single)
Rios et al. (2022)	Series (retrospective)	22	27.4	Bimaxillary osteotomy	Maxilla (segmental)
Ruckschloss et al. (2020)	Cohort (retrospective)	29	24.2	Bimaxillary osteotomy	Chin
Ruckschloss et al. (2019)	Cohort (retrospective)	18	F: 23.3 M: 19.8	Le Fort I osteotomy (single) or bimaxillary osteotomy	Maxilla (single)
Sanchez-Jauregui et al. (2020)	Cohort (prospective)	30	NS	Le Fort I osteotomy (single or split)	Maxilla (single or segmental)
van der Wel et al. (2022)	RCMCT (prospective)	27 ^†^	PSI: 28.6 (9.7) Splint: 26.8 (6.9)	Le Fort I osteotomy (single) with or without mandibular osteotomy	Maxilla (single)
Wong et al. (2021)	Series (retrospective)	30	F: 25.5 M: 29.2	Le Fort I osteotomy (single) with or without mandibular osteotomy and genioplasty	Maxilla (single)

* Same cohort as Badiali et al. (2020); RCT: randomized controlled trial; RCMCT: randomized controlled multi-center trial; PSI; patient-specific implant; F/M: female/male; NS: not specified; † sub-group of cohort in Kraeima et al. (2020).

**Table 2 jpm-13-00125-t002:** Accuracy measurements.

Author, Year	PSI Component (s)	Translation	Rotation
Abel et al. (2022)	Maxilla and mandible	A-point: Lateral: 0.37 (0.45), AP: 0.57 (0.57), vertical: 0.45 (0.42) B-point: Lateral: 0.62 (0.54), AP: 1.15 (1.28), vertical: 1.14 (0.83) Pogonion: Lateral: 0.85 (0.73), AP: 1.29 (1.50), vertical: 1.24 (0.85)	
Badiali et al. (2021)	Mandible	Mandible: Lateral: 1.14 (1.00), AP: 1.49 (1.59), vertical: −1.05 (1.23)	Mandible: Pitch: 1.53 (1.34), roll: 0.96 (0.85), yaw: 1.13 (0.99)
Badiali et al. (2020)	Mandible	Rami: Lateral: 0.49 (1.05), AP: 0.32 (0.92), vertical: −0.04 (0.94) Plates: Lateral: 0.20 (0.64), AP: −0.68 (1.18), vertical: 0.26 (0.79)	Rami: Pitch: 0.52 (2.39), roll: 0.90 (2.15), yaw: −1.91 (2.70) Plates: Pitch: −1.93 (3.89), roll: 0.13 (3.04), yaw: 0.20 (1.41)
Greenberg et al. (2021)	Maxilla	UI: Lateral: 0.66 (0.51), AP: 0.80 (0.56), vertical: 0.48 (0.27)	
Hanafy et al. (2020)	Maxilla	PSI UI: Lateral: 0.07 (0.02), AP: 0.17 (0.04), vertical: 0.26 (0.08) Wafer UI: Lateral: 0.71 (0.21), AP: 1.31 (0.51), vertical: 1.45 (0.58)	
Harding et al. (2022)	Mandible	R ramus PSI: Lateral: 1.18, AP: 0.62, vertical: 0.58 R ramus control: Lateral: 0.83, AP: 0.78, vertical: 0.70 L ramus PSI: Lateral: 1.20, AP: 0.45, vertical: 0.44 L ramus control: Lateral: 0.83, AP: 0.68, vertical: 0.56	R ramus PSI: Pitch: 1.00, roll: 1.43, yaw: 1.53 R ramus control: Pitch: 1.50, roll: 2.42, yaw: 2.98 L ramus PSI: Pitch: 1.32, roll: 1.62, yaw: 1.59 L ramus control: Pitch: 1.41, roll: 1.39, yaw: 2.45
Heufelder et al. (2017)	Maxilla	UI: Lateral: 0.2, AP: 0.35, vertical: 0.85	
Ho et al. (2020)	Maxilla	Maxilla: Lateral: 0.3, AP: 1.1, vertical: 0.5	Maxilla: Pitch: 1.9, roll: 0.4, yaw: 0.1
Jones et al. (2022)	Maxilla	PSI UI: Lateral: 0.41, AP: 0.94, vertical: 0.54 Splint UI: Lateral: 1.01, AP: 0.93, vertical: 1.23	
Karanxha et al. (2021)	Maxilla and mandible	PSI UI: Lateral: 0.32 (0.27), AP: 1.43 (0.81), vertical: 0.85 (0.59) Splint UI: Lateral: 0.45 (0.43), AP: 1.53 (0.63), vertical: 1.73 (0.90) PSI LI: Lateral: 0.94 (0.74), AP: 1.99 (1.84), vertical: 1.99 (1.39) Splint LI: Lateral: 1.78 (1.11), AP: 1.34 (1.2), vertical: 3.68 (4.62)	PSI Maxilla: Pitch: 2.87 (1.52), roll: 0.49 (0.51), yaw: 0.17 (0.05) Splint maxilla: Pitch: 2.30 (1.31), roll: 1.62 (0.78), yaw: 0.63 (0.45) PSI mandible: Pitch: 2.85 (1.68), roll: 0.84 (0.82), yaw: 1.68 (1.00) Splint mandible: Pitch: 2.48 (2.30), roll: 1.54 (1.85), yaw: 1.62 (1.36)
Kraeima et al. (2020)	Maxilla	PSI UI: Lateral: 0.46, AP: 1.05, vertical: 0.87 Splint UI: Lateral: 1.07, AP: 1.74, vertical: 0.98	PSI Maxilla: Pitch: 2.33, roll: 0.53, yaw: 0.21 Splint maxilla: Pitch: 2.17, roll: 0.60, yaw: 0.44
Li et al. (2017)	Maxilla and mandible	UI: Lateral: −0.18 (0.35), AP: −0.54 (0.53), vertical: 0.33 (0.53) LI: Lateral: −0.33 (0.50), AP: −0.67 (0.92), vertical: 0.38 (0.72) Left ramus: Lateral: −0.10 (1.03), AP: 0.23 (0.82), vertical: −0.10 (0.79) Right ramus: Lateral: −0.18 (0.70), AP: 0.05 (0.54), vertical: −0.28 (0.94)	UI: Pitch: 0.11 (0.69), roll: 0.02 (0.60), yaw: −0.83 (1.83) LI: Pitch: 0.45 (1.67), roll: −0.07 (0.95), yaw: 0.26 (0.96) Left ramus: Pitch: 1.39 (2.12), roll: 0.01 (1.14), yaw: 0.49 (2.06) Right ramus: Pitch: −1.66 (1.85), roll: −0.59 (1.73), yaw: 0.26 (2.20)
Li et al. (2020)	Chin	Chin centroid: Lateral: 0.06 (0.71), AP: −0.49 (0.46), vertical: 0.39 (0.55)	Chin centroid: Pitch: 0.68 (1.68), roll: −0.10 (1.67), yaw: −0.17 (2.08)
Li et al. (2021)	Maxilla	PSI maxilla centroid: Lateral: 0.37 (0.40), AP: 1.02 (0.66), vertical: 0.61 (0.44) Splint maxilla centroid: Lateral: 1.23 (0.93), AP: 1.12 (0.82), vertical: 0.96 (0.74)	PSI maxilla: Pitch: 1.85 (1.42), roll: 1.63 (0.52), yaw: 0.63 (0.44) Splint maxilla: Pitch: 1.72 (1.56), roll: 1.25 (1.18), yaw: 0.88 (0.71)
Liu et al. (2020)	Maxilla	PSI: Lateral: 0.30 (0.18), AP: 0.50 (0.18), vertical: 0.33 (0.18) Splint: Lateral: 1.05 (0.75), AP: 1.37 (0.84), vertical: 1.32 (1.40)	PSI: Pitch: 0.40 (0.31), roll: 0.28 (0.21), yaw: 0.82 (0.60) Splint: Pitch: 3.56 (3.26), roll: 1.53 (1.05), yaw: 8.84 (7.73)
Rios et al. (2022)	Maxilla	UI: Lateral: 0.54 (0.44), AP: 0.74 (0.51), vertical: 0.35 (0.24)	
Ruckschloss et al. (2020)	Chin	Chin: Lateral: 0.25 (0.28), AP: 0.70 (0.64), vertical: 0.45 (0.38)	Chin: Pitch: 1.76 (0.98), roll: 0.89 (0.74), yaw: 0.83 (0.57)
Ruckschloss et al. (2019)	Maxilla	PSI: Lateral: 0.51 (0.48), AP: 0.39 (0.26), vertical: 0.37 (0.40) Splint: Lateral: 1.11 (1.32), AP: 1.42 (0.87), vertical: 0.62 (0.47)	PSI: Pitch: 1.04 (0.97), roll: 0.43 (0.49), yaw: 0.57 (0.26) Splint: Pitch: 1.84 (1.48), roll: 0.46 (0.64), yaw: 1.31 (1.11)
Sanchez-Jauregui et al. (2020)	Maxilla	PSI: Lateral: 0.2, AP: 0.8, vertical: 0.4 Splint: Lateral: 0.2, AP: 1.7, vertical: 1.8	
van der Wel et al. (2022)	Maxilla	Accuracy: PSI UI: Lateral: 0.5, AP: 3.8, vertical: 2.1 Splint UI: Lateral: 1.2, AP: 2.8, vertical: 1.5 Stability: PSI UI: Lateral: 0.3, AP: 0.5, vertical: 0.3 Splint UI: Lateral: 0.2, AP: 0.2, vertical: 0.3	Accuracy: PSI: Pitch: 2.6, roll: 0.7, yaw: 0.2 Splint: Pitch: 1.8, roll: 0.9, yaw: 0.6 Stability: PSI: Pitch: 0.1, roll: 0.2, yaw: 0.0 Splint: Pitch: 0.0, roll: 0.2, yaw: 0.0
Wong et al. (2021)	Maxilla	UI: Lateral: −0.14 (0.22), AP: −0.09 (0.38), vertical: −0.26 (0.32)	UI: Pitch: −0.28 (1.03), roll: 0.09 (0.47), yaw: −0.21 (0.45)

UI: upper incisor midpoint; PSI: patient-specific implant; R: right; L: left.

**Table 3 jpm-13-00125-t003:** Risk of bias analysis using the revised methodological items for non-randomized studies [26].

Record	MINORS Score
Karanxha et al. (2021)	18
Harding et al. (2022)	17
Jones et al. (2022)	16
Liu et al. (2020)	16
Ruckschloss et al. (2019)	16
Heufelder et al. (2017)	12
Li et al. (2020)	12
Abel et al. (2022)	10
Ho et al. (2020)	10
Li et al. (2017)	10
Rios et al. (2022)	10
Wong et al. (2021)	10
Badiali et al. (2021)	9
Badiali et al. (2020)	9
Greenberg et al. (2021)	9
Kim et al. (2019)	9
Ruckschloss et al. (2020)	8

**Table 4 jpm-13-00125-t004:** Risk of bias analysis using Cochrane Collaboration’s risk of bias tool.

Record	1	2	3	4	5	6
Hanafy et al. (2020)	+	+	+	+	+	+
Li et al. (2021)	+	+	+	+	+	+
Sanchez-Jauregui et al. (2020)	+	?	?	?	+	+
Kraeima et al. (2020)	+	?	?	?	+	+
van der Wel et al. (2022)	+	?	?	?	- *	+

1. Random sequence generation; 2. Allocation concealment; 3. Blinding of participants; 4. Blinding of outcome; 5. Incomplete outcome data addressed; 6. Selective reporting; +: low risk; ?: unclear risk; -: high risk; * sub-group of Kraeima et al. (2020) cohort: new time-point of outcome (1 year postoperatively).

**Table 5 jpm-13-00125-t005:** Descriptive results on translational error and significance at 2 mm threshold.

Summarized Results	Lateral			Antero-Posterior			Vertical		
	Mean	SD	*p* *	Mean	SD	*p* *	Mean	SD	*p* *
Absolute values	0.30	0.18	<0.0001	0.50	0.13	<0.0001	0.42	0.14	<0.0001
Hanafy et al.	0.07	0.02	<0.0001	0.17	0.04	<0.0001	0.26	0.08	<0.0001
Signed values	−0.17	0.41	<0.0001	−0.24	0.31	<0.0001	−0.10	0.28	<0.0001
Overall	0.07	0.02	<0.0001	0.30	0.08	<0.0001	0.28	0.07	<0.0001

* simple one-sample *t*-test for the estimated mean <2 mm.

**Table 6 jpm-13-00125-t006:** Descriptive results on rotational error and significance at 4° threshold.

Summarized Results	Pitch			Roll			Yaw		
	Mean	SD	*p* *	Mean	SD	*p* *	Mean	SD	*p* *
Absolute values	0.65	0.35	<0.0001	0.36	0.17	<0.0001	0.26	0.12	<0.0001
Signed values	−0.01	0.57	<0.0001	0.06	0.37	<0.0001	−0.24	0.44	<0.0001
Overall	0.45	0.25	<0.0001	0.15	0.15	<0.0001	0.10	0.10	<0.0001

* simple one-sample *t*-test for the estimated mean <4°.

**Table 7 jpm-13-00125-t007:** Translational differences and significance between PSI and conventional osteosynthesis.

Dimension	Mean Difference (mm)	*p*-Value *
Lateral	Absolute values only: −0.852	Absolute values only: <0.0001
	Overall: −0.852	Overall: <0.0001
Antero-posterior	Absolute values only: −0.677	Absolute values only: <0.0001
	Overall: −1.127	Overall: <0.0001
Vertical	Absolute values only: −0.393	Absolute values only: <0.0001
	Overall: −0.811	Overall: <0.0001

* two-sample *t*-test.

**Table 8 jpm-13-00125-t008:** Rotational differences and significance between PSI and conventional osteosynthesis.

Dimension	Mean Difference (Degrees)	*p*-Value *
Pitch	−2.352	<0.0001
Roll	−0.472	<0.0001
Yaw	−0.568	<0.0001

* two-sample *t*-test.

## Data Availability

Not applicable.

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
