# Peer review of "Three-Dimensional Accuracy and Stability of Personalized Implants in Orthognathic Surgery: A Systematic Review and a Meta-Analysis"

_jpm, 2023, doi:10.3390/jpm13010125_

Round 1

Reviewer 1 Report

The current paper is a review with meta-analysis that compared the accuracy and stability of PSI versus conventional CAD/CAM in OS.

The methodologies used (PRISMA and MINORS) were appropriate for the study. The references were exhaustively used and discussed.

If there is one point of improvement, the authors may look into replacing reference no. 8 (Maes, 1997) if deemed appropriate.

Author Response

We thank the reviewer for the kind words and for taking the time to review our submission.

Regarding reference 8. Maes, F.; Collignon, A.; Vandermeulen, D.; Marchal, G.; Suetens, P. Multimodality image registration by maximization of mutual information. IEEE Trans Med Imaging 1997, 16, 187-198, doi:10.1109/42.563664. We strongly urge to keep this reference, as voxel-based registration was introduced in this work.

Reviewer 2 Report

my compliments for your job and to clarify the situation

Probably you have to spend some words about ICP algorithm that provides a reproducible method of comparison and alignment between 3D models.

Three-Dimensional Comparison of the Maxillary Surfaces through ICP-Type Algorithm: Accuracy Evaluation of CAD/CAM Technologies in Orthognathic Surgery

Cassoni, A.Manganiello, L.Barbera, G., ...Pucci, R.Valentini, V. International Journal of Environmental Research and Public Healththis link is disabled2022, 19(18), 11834   Rückschloß T., Ristow O., Müller M., Kühle R., Zingler S., Engel M., Hoffmann J., Freudlsperger C. Accuracy of patient-specific implants and additive-manufactured surgical splints in orthognathic surgery—A three-dimensional retrospective study. J. Cranio-Maxillofac. Surg. 2019;47:847–853. doi: 10.1016/j.jcms.2019.02.011.   Marlière D.A.A., Demétrio M.S., Verner F.S., Asprino L., Chaves Netto H.D.D.M. Feasibility of iterative closest point algorithm for accuracy between virtual surgical planning and orthognathic surgery outcomes. J. Cranio-Maxillofac. Surg. 2019;47:1031–1040. doi: 10.1016/j.jcms.2019.03.025.

Author Response

We thank the reviewer for the kind words and for taking the time to review our submission.

We thank the reviewer for the suggestion to include a discussion on the ICP algorithm, a.k.a. surface-based registration. We agree that both surface- and voxel-based registration has been proven to be reproducible methods for 3D assessment of the surgical accuracy, and it is an important point of discussion. To accommodate this suggestion, we have added a paragraph on ICP (surface-based registration), including the proposed references, and its comparison to voxel-based registration for 3D assessment of the accuracy and stability of orthognathic surgery:

“It should be noted that surface-based registration (SBR) provides an alternative method to VBR for 3D assessment of the surgical accuracy and stability of orthognathic surgery, which has been applied in several of the included studies in the present systematic literature review [13,29-32,35,36,40,43,45-48]. Surface-based registration aligns two 3D surfaces using the iterative closest point algorithm [49], and SBR has shown to be a reproducible method [50,51]. Both registration methods have been found to be reliable and accurate [52-54]. Although VBR has been shown to be more consistent and efficient than SBR, the differences between the two methods were statistically insignificant [52-54]. Another comparative study proved that SBR was more accurate and reliable than VBR on the mandibular ramus for long-term 3D assessment of condylar remodeling following orthognathic surgery [55]. However, it was concluded that the performance difference might have been caused by the application of an inappropriate reference structure proposed in the literature [55,56]. Hence, according to the literature, no statistically significant nor clinical relevant difference have yet been shown between the performances of the two methods. Consequently, both methods have been found applicable for 3D assessment of the surgical accuracy and stability of orthognathic surgery.”

  1. Ruckschloss, T.; Ristow, O.; Kuhle, R.; Weichel, F.; Roser, C.; Aurin, K.; Engel, M.; Hoffmann, J.; Freudlsperger, C. Accuracy of laser-melted patient-specific implants in genioplasty - A three-dimensional retrospective study. Journal of Cranio-Maxillofacial Surgery 2020, 48(7), 653-660, doi:http://dx.doi.org/10.1016/j.jcms.2020.05.003.
  2. Besl, P.J.; McKay, N.D. A method for registration of 3-D shapes. Ieee T Pattern Anal 1992, 14, 239-256, doi:10.1109/34.121791.
  3. Cassoni, A.; Manganiello, L.; Barbera, G.; Priore, P.; Fadda, M.T.; Pucci, R.; Valentini, V. Three-Dimensional Comparison of the Maxillary Surfaces through ICP-Type Algorithm: Accuracy Evaluation of CAD/CAM Technologies in Orthognathic Surgery. Int J Environ Res Public Health 2022, 19, doi:10.3390/ijerph191811834.
  4. Marliere, D.A.A.; Demetrio, M.S.; Verner, F.S.; Asprino, L.; Chaves Netto, H.D.M. Feasibility of iterative closest point algorithm for accuracy between virtual surgical planning and orthognathic surgery outcomes. J Craniomaxillofac Surg 2019, 47, 1031-1040, doi:10.1016/j.jcms.2019.03.025.
  5. Han, G.; Li, J.; Wang, S.; Wang, L.; Zhou, Y.; Liu, Y. A comparison of voxel- and surface-based cone-beam computed tomography mandibular superimposition in adult orthodontic patients. J Int Med Res 2021, 49, 300060520982708, doi:10.1177/0300060520982708.
  6. Ghoneima, A.; Cho, H.; Farouk, K.; Kula, K. Accuracy and reliability of landmark-based, surface-based and voxel-based 3D cone-beam computed tomography superimposition methods. Orthod Craniofac Res 2017, 20, 227-236, doi:10.1111/ocr.12205.
  7. Almukhtar, A.; Ju, X.; Khambay, B.; McDonald, J.; Ayoub, A. Comparison of the accuracy of voxel based registration and surface based registration for 3D assessment of surgical change following orthognathic surgery. PLoS One 2014, 9, e93402, doi:10.1371/journal.pone.0093402.
  8. Holte, M.B.; Saederup, H.; Pinholt, E.M. Comparison of surface- and voxel-based registration on the mandibular ramus for long-term three-dimensional assessment of condylar remodelling following orthognathic surgery. Dentomaxillofac Radiol 2022, 51, 20210499, doi:10.1259/dmfr.20210499.
  9. Verhelst, P.J.; Shaheen, E.; de Faria Vasconcelos, K.; Van der Cruyssen, F.; Shujaat, S.; Coudyzer, W.; Salmon, B.; Swennen, G.; Politis, C.; Jacobs, R. Validation of a 3D CBCT-based protocol for the follow-up of mandibular condyle remodeling. Dentomaxillofac Radiol 2020, 49, 20190364, doi:10.1259/dmfr.20190364.